# Protective Effects of Natural Antioxidants on Inflammatory Bowel Disease: Thymol and Its Pharmacological Properties

**DOI:** 10.3390/antiox11101947

**Published:** 2022-09-29

**Authors:** Yao Liu, Hui Yan, Bing Yu, Jun He, Xiangbing Mao, Jie Yu, Ping Zheng, Zhiqing Huang, Yuheng Luo, Junqiu Luo, Aimin Wu, Daiwen Chen

**Affiliations:** Key Laboratory for Animal Disease-Resistance Nutrition of China Ministry of Education, China Ministry of Agriculture and Rural Affairs and Sichuan Province, Animal Nutrition Institute, Sichuan Agricultural University, Chengdu 611130, China

**Keywords:** gastrointestinal disease, gut health, natural anti-oxidant, oxidative stress, chronic inflammation

## Abstract

Inflammatory bowel disease (IBD) is a gastrointestinal disease that involves chronic mucosal or submucosal lesions that affect tissue integrity. Although IBD is not life-threatening, it sometimes causes severe complications, such as colon cancer. The exact etiology of IBD remains unclear, but several risk factors, such as pathogen infection, stress, diet, age, and genetics, have been involved in the occurrence and aggravation of IBD. Immune system malfunction with the over-production of inflammatory cytokines and associated oxidative stress are the hallmarks of IBD. Dietary intervention and medical treatment suppressing abnormal inflammation and oxidative stress are recommended as potential therapies. Thymol, a natural monoterpene phenol that is mostly found in thyme, exhibits multiple biological functions as a potential adjuvant for IBD. The purpose of this review is to summarize current findings on the protective effect of thymol on intestinal health in the context of specific animal models of IBD, describe the role of thymol in the modulation of inflammation, oxidative stress, and gut microbiota against gastrointestinal disease, and discuss the potential mechanism for its pharmacological activity.

## 1. Introduction

Inflammatory bowel disease (IBD) is a chronic and relapsing inflammatory disorder of the gastrointestinal tract (GIT) [1]. It is not fatal, but imposes a burden on global healthcare and greatly reduces the quality of life of patients. From 1990 to 2017, the number of IBD patients increased from 37 million to more than 68 million, and the global prevalence of IBD increased by 85.1% [2]. The United States has the highest age-standardized prevalence rate in the world, accounting for nearly a quarter of all global IBD patients in 2017. Among European countries, the United Kingdom has the highest age-standardized prevalence. The prevalence of IBD in the United States ranges from 252 to 439 cases per 100,000 people [3]. Compared with western countries, the incidence of IBD in Asia is relatively low, but it has risen from 0.54 to 3.44 per 100,000 individuals across eight Asian countries [4,5]. The prevalence and incidence of IBD are continuing to increase worldwide in different races and countries [6].

IBD includes Crohn’s disease (CD) and ulcerative colitis (UC). UC involves long-lasting inflammation and ulcers along the superficial lining of the large intestine (rectum and colon), while CD is characterized by the discontinuous transmural inflammation in deeper layers of most regions of the GIT [7,8]. The exact cause of IBD remains unknown, but malfunction in the immune system is its hallmark. In IBD patients, an abnormal immune response to pathogen infection or stress causes chronic inflammation in the GIT contributing to the incidence of the disease [9]. Some genetic factors involved in an inappropriate immune response are a reason for IBD [10]. Oxidative stress (OS) with excessive accumulation of reactive oxygen species (ROS) has been reported to correlate with intestinal chronic inflammation and incidence of IBD [11]. Moreover, alteration in the gut microbiota has been associated with the pathogenesis of IBD [12]. Unfortunately, there is currently no cure for IBD. Anti-inflammation medicines are used to relieve inflammation and even surgery is required to remove damaged parts of the GIT when patients suffer. Recently, advances in dietary regulation of inflammation, OS, and gut microbiota have resulted in dietary intervention against IBD incidence being recommended as a long-term prevention and therapy technique [13]. Currently, more research is focusing on natural plant extracts, such as phenol compounds, for the prevention and relief of IBD [14].

Thymol (2-isopropyl-5-methylphenol), a natural monoterpene phenol compound, is the major component of the essential oils extracted from plants of thyme species, such as *Thymus vulgaris*, *Coridothymus capitatus*, and *Origanum vulgare* [15]. Thymol exhibits multiple biological and pharmacological properties, including anti-inflammation, anti-oxidation, anti-bacteria, anti-fungal, and anti-tumor potential [16]. A high dose of thymol up to 500 mg/kg diet has been shown to have no toxicity [17]. Thus, thymol is considered as a beneficial food supplement. Recent studies reported that thymol improves intestinal integrity and alleviates intestinal injury via the regulation of the immune response and oxidation-reduction homeostasis [18]. In this review, we discussed the potential of thymol as a natural anti-inflammatory, anti-oxidative, and anti-bacterial compound in the modulation of mucosal immunity, OS, and gut microbes for IBD treatment.

## 2. Pharmacokinetics and Pharmacological Properties of Thymol

Thymol is the secondary metabolite produced by the aromatization of γ-terpinene to *p*-cymene, followed by the hydroxylation of *p*-cymene in plants [19] (Figure 1), including *Thymus zygis* [20], *Thymbra capitata* [21], *Thymus vulgaris* [22], *Satureja thymbra* [23], *Nigella sativa* seeds [24], and *Monarda didyma* [25]. Pure thymol has low solubility in water, high volatility, and a strong bitter/irritating taste [26]; thus, it is usually encapsulated in electrospun nanofibers to enhance its water solubility and high temperature stability [27]. Emulsification is also used for thymol processing. Both encapsulation and emulsification have been shown to improve the anti-oxidant activity of thymol [28].

Thymol administered orally can be rapidly absorbed in the stomach and small intestines, and then transported to various organs via the circulation system in animals [29]. Thymol can be metabolized to thymol sulfate and thymol glucuronide by sulfation and glucuronidation in the intestines, liver, kidney, and other organs, respectively [30]. Thymol and its metabolites reach the maximum concentration in the blood 30 min after oral administration and then are slowly eliminated in about 24 h [26]. They also present in the lungs, kidneys, mucosa of the small and large intestines, and other organs, suggesting that they may function directly in these organs. It is difficult to determine which compound is the active form of thymol, because thymol metabolites can be deconjugated to thymol locally and export its pharmacological activity in this way [31]. Recently, scientists have focused on the pharmacological activities of thymol.

Thymol-containing plants have long been used in traditional Chinese medicine due to its pharmacological properties. Thymol acts as a potent inhibitor of the release of inflammatory cytokines, such as interleukin (IL)-6, IL-1β, and IL-8 [32]. Thymol enhances anti-oxidative capacity to alleviate OS in different tissues [33]. As a spectrum anti-bacterial agent, thymol reduces the activities of microorganisms belonging to the Enterobacteriaceae, Streptococcus, and Saccharomycetaceae families, involving membrane rupture, inhibition of biofilm formation, and other pathways [34,35]. Thymol shows anti-viral potential to inhibit virus colonization, such as severe acute respiratory syndrome coronavirus 2 (SARS-CoV-298), by docking with the S1 receptor-binding domain of spike glycoprotein [36]. Thymol also exhibits other pharmacological functions, such as analgesia, anti-malarial, and anti-fungal properties [37,38,39,40,41]. This review focuses on the therapeutic potential of thymol in IBD in terms of its pharmacological properties and discusses the underlying mechanisms.

## 3. Thymol Protects Intestinal Barrier Function against IBD

The intestinal barrier, mainly composed of intestinal epithelial cells (IECs) and immune cells, maintains the balance between the luminal contents and mucosa. The disturbance of this balance has been associated with gastrointestinal diseases, such as IBD [42]. Although the exact pathogenesis of IBD remains unclear, a “leaky gut” with impaired intestinal barrier function is the main feature. The intestinal barrier is the first line of defense against pathogen infection, and injury to the intestinal barrier aggravates the disease. Thus, understanding how thymol protects the intestinal barrier is important for relieving IBD.

Thymol has exhibited a protective function for the intestinal barrier in both in vivo and in vitro studies, as illustrated in Table 1, and its specific mechanism is shown in Figure 2. Thymol attenuates weaning stress-induced diarrheal and intestinal barrier dysfunction in weanling pigs by reducing the serum diamine oxidase level, an indicator of intestinal integrity, and increasing the expression of the tight junction protein zonula occludens-1 (ZO-1) and occludins [43]. Thymol alleviates dextran sulfate sodium (DSS)-induced intestinal damage and increases tight junction claudin-3 expression [44]. Increased plasma endotoxin and D-lactic acid levels are markers of increased intestinal permeability. The latest study found that dietary thymol reduced the plasma endotoxin and D-lactic acid concentrations on days 7 and 14 post-weaning [45]. The intestinal mucus layer is the first line of defense maintaining bacterial symbiosis with the host and preventing bacterial penetration into epithelial cells [46]. Thymol increases mucus secretion to relieve ethanol-induced ulcer mucosal damage in rats [47]. In IPEC-J2 cells, thymol alleviates lipopolysaccharide (LPS)-induced decrease in trans-epithelial electrical resistance (TEER), indicating an increase in the integrity of the single cell layer [48]. In Caco-2 cells, thymol increases the integrity of the tight junction and up-regulates cyclooxygenase-1 (COX1) activity to maintain GIT homeostasis, which is beneficial for intestinal health [49]. In addition, thymol changes the expressions of 120 and 59 genes in the oxyntic and pyloric mucosa, respectively, which are associated with gastric epithelium proliferation and maturation activities in weaned pigs [29].

## 4. Thymol Alleviates Intestinal Inflammation in IBD

The dysregulation of innate and adaptive immune responses leads to chronic intestinal inflammation in IBD patients [51,52,53]. Transcription factor nuclear factor κB (NF-κB) is the key mediator regulating the inflammatory response. The activation of NF-κB signaling induces the expression of pro-inflammatory cytokines, including tumor necrosis factor-α (TNF-α), inductible nitric oxide synthase (iNOS), interleukin (IL-1β), and IL-6 [54]. The over-production and accumulation of inflammatory cytokines cause intestinal epithelial apoptosis and the disruption of intestinal homeostasis, resulting in the dysfunction of the intestinal epithelial barrier in IBD patients [55]. Currently, the suppression of inflammation is the mainstay of IBD treatment.

Thymol shows strong anti-inflammatory properties in in vivo and in vitro studies [56]. The anti-inflammatory properties of thymol are listed in Table 2 and a schematic diagram of thymol’s action is depicted in Figure 3. Toll-like receptors (TLRs) are the main sensors used to detect various dangerous signals and activate innate immune responses. The classic TLR signaling pathway activates NF-κB to regulate the expressions of a series of cytokines [57]. In mice and macrophages, thymol inhibits TLR4 expression and then inhibits the activation of NF-κB signaling, which reduces the production of inflammatory cytokines, such as TNF-α and IL-1β [58,59]. NF-κB is a master mediator of inflammatory responses. Inactive NF-κB binds to IκB, an inhibitory subunit of NF-κB, and presents in the cytoplasm. When activated by a variety of signals, such as cytokine receptors and pattern-recognition receptors (PRRs), IκB is phosphorylated and degraded to release RelA (p65) from the NF-κB complex. p65 is then translocated to the nucleus to induce pro-inflammatory cytokine expression as a transcription factor [60]. Thymol has been shown to inhibit NF-κB activation by reducing p65 translocation and abundance in the colons of acetic acid-induced colitis rats and in LPS-activated macrophages, respectively, with decreased cytokine production [61,62]. The activation of NF-κB induces the expressions of iNOS and COX-2, which further promote vigorous inflammation. In ulcerative colitis rats, thymol reduces the COX-2 expression and nitric oxide (NO) levels produced by iNOS in the rats’ colon [63]. These studies showed the anti-inflammatory function of thymol through the inhibition of the NF-κB signaling pathway.

Studies have reported that proteins in the MAPK family, such as JNK1/2, p38α, and ERK, are activated in the colonic mucosa of IBD patients [71]. The suppression of the MAPK signaling pathway is an approach for alleviating inflammation and thus IBD [71]. It has been reported that thymol inhibits p38 phosphorylation and interferes with the activation of the MAPK signaling pathway to maintain the immune balance [64]. Thymol also suppresses LPS-induced activation of p-p38, p-JNK, and p-ERK, and correspondingly inhibits the production of NO, IL-6, TNF-α, COX-2, and other inflammatory cytokines [72]. Therefore, thymol can also reduce the inflammatory response by inhibiting the MAPK signaling pathway.

Cytokines are synthesized and secreted by activated immune cells, such as macrophages, T cells, B cells, dendritic cells (DCs), and natural killer cells. Among them, the regulatory T cell (Treg) is essential to control autoimmunity. Treg is defined by the expression of CD4, CD25, and transcription factor forkhead box P3 (Foxp3) [73]. Thymol promotes the differentiation of naïve T cells to CD4^+^CD25^+^Foxp3^+^ Treg cells and induces Foxp3 expression [74]. Meanwhile, thymol also maintains the balance of the Th1/Tregs and Th17/Tregs ratios to prevent autoimmunity as a result of suppressed inflammation [74,75]. Additionally, thymol exerts inhibitory effects on DCs’ maturation and T cell activation [76]. Although thymol has influenced the immune cell population in some studies, more information is required on how thymol modulates immune cell differentiation and whether the changes in the immune cell population by thymol contribute to the relief of IBD.

## 5. Thymol Improves Anti-Oxidant Capacity in IBD

IBD has been also associated with intestinal OS [77]. OS is the result of the excessive accumulation of ROS caused by the imbalance of the oxidation and anti-oxidation systems in cells [78]. Generally, ROS include hydrogen peroxide (H_2_O_2_), superoxide anions (O_2_^•−^), and hydroxyl radicals (HO•). The accumulation of ROS alters the structure and function of cell contents, such as DNA and proteins, and causes a series of cellular dysfunctions, including the disruption of cell metabolism, disruption of cell cycles, dysregulation of the immune response, etc. In IBD patients, the overproduction of ROS destroys the cytoskeleton and interrupts tight junction proteins in the intestinal epithelium, resulting in increased epithelial permeability and intestinal barrier dysfunction [79]. In addition, OS has been reported to be positively correlated with the level of inflammation [80], and OS in the GIT exacerbates intestinal inflammation and IBD. Therefore, the relief of intestinal OS is another strategy for treating IBD [81].

Thymol has been shown to have anti-oxidative capacity in a variety of OS models in vitro and in vivo, as listed in Table 3. A schematic mechanism of thymol’s action is depicted in Figure 4. Thymol acts as a strong anti-oxidant. partially due to its phenolic hydroxyl groups, which directly neutralize free radicals [82]. 2,2-diphenyl-1-picrylhydrazyl (DPPH) and 2,2′-azino-bis (3-ethylbenzothiazoline-6-sulfonic acid) (ABTS^+^) are commonly used free radicals to evaluate anti-oxidant activity. Thymol eliminates 50% of DPPH radicals and scavenges ABTS^+^ radicals as a direct anti-oxidant in vitro [83]. However, other studies have indicated that phenolic compounds exert an anti-oxidative function through the activation of anti-oxidation-related signaling pathways, rather than directly neutralizing free radicals by the phenolic hydroxyl group [84]. Studies indicated that dietary polyphenols activate the anti-oxidation-related nuclear factor-E2-related factor 2 (Nrf2) signaling pathway, which plays a pivotal role in regulating the expression of anti-oxidant genes and the activities of anti-oxidant enzymes [85]. Studies have indicated that thymol activates Nrf2 signaling in different tissues [86,87]. Importantly, the Nrf2 downstream target HO-1 catalyzes the degradation of heme into Fe^2+^, CO, and bilirubin. Heme enhances ROS formation in OS, while bilirubin is an anti-oxidant and protects tissues [88]. Thymol promotes HO-1 expression in the lungs of mice, accounting for its anti-oxidative capacity [87]. Moreover, HO-1-induced CO also performs anti-inflammation and anti-apoptosis functions, which support the mechanism of thymol alleviating inflammation and intestinal damage.

Nrf2 maintains the cellular redox balance and prevents oxidative damage by regulating anti-oxidant enzymes, such as GSH and GPX. As a result, thymol increases the activities of anti-oxidant enzymes to reduce ROS production. SOD reduces superoxide O_2_^•−^ radicals by catalyzing them into O_2_ and H_2_O_2_. H_2_O_2_ is then transformed by either CAT or GPX in the peroxisome to generate water and O_2_. Thymol enhances the activities of anti-oxidant enzymes and alleviates OS in animal models of OS [89]. Thymol enhances the total anti-oxidative capacity and reduces the ROS level in rats challenged with imidacloprid [93]. Thymol reduces the MDA level, a byproduct of lipid peroxidation and a marker of OS, increases SOD activity, and ameliorates OS in mouse obesity models [95]. In in vitro studies, thymol ameliorates acetaminophen-induced OS in HepG2 cells by increasing the GPX and SOD activities and decreasing the MDA level [99]. In lung fibroblasts, thymol increases the enzyme activities of SOD, CAT, and GPX, leading to a reduction in ROS production [90]. In the Chang liver cell line, thymol inhibits ROS production by increasing the GPX level and decreasing the MDA level, alleviating t-butyl-hydroperoxide-induced oxidative damage. In addition to the aforementioned anti-oxidant enzymes, thymol reduces OS through the inhibition of free-radical-generating enzyme xanthine oxidase (XO) [100]. XO catalyzes the oxidation of xanthine and hypoxanthine with increased production of O_2_^•−^ and H_2_O_2_ in purine metabolism [101]. Thymol binds to XO directly and inhibits XO activity to produce ROS [100]. Collectively, thymol relieves OS via directly scavenging free radicals by its phenolic hydroxyl group, directly inhibiting free-radical-producing enzymes, and indirectly activating the Nrf2-mediated signaling pathway.

## 6. Thymol Changes Gut Microbes and Prevents Pathogen Infection

Mammalian gut microbes are mainly composed of four phyla, including Firmicutes, Bacteroidetes, Proteobacteria, and Actinobacteria. Alterations in the gut microbiota are observed in patients with IBD compared with healthy individuals. Notably, the abundance of the phylum Firmicutes is reduced in the stool of UC patients [102]. Although there is a lack of clinical research about the thymol-modulated gut microbial structure in IBD, multiple animal studies have described increases in the proportional abundance of Firmicutes potentially against IBD pathogenesis due to thymol [103]. However, a direct causative relationship between IBD and dysbiosis has not been clearly established in humans. The change in gut microbes by thymol as a therapeutical mechanism requires more elucidative studies.

Currently, studies of the microbial etiology of IBD indicate that persistent pathogen infection, such as enterotoxic *E. coli*, causes a “leaky gut” and chronic inflammation, subsequently leading to the excessive translocation of intestinal bacteria and the dysbiosis between “beneficial” and “detrimental” bacteria, resulting in IBD [104]. The members of the Proteobacteria phylum, notably the Enterobacteriaceae *E. coli*, are increased in IBD patients relative to healthy individuals [105]. Several studies have indicated the anti-pathogen capacity of thymol in different animal infection models and bacterial cultures, as illustrated in Table 4, as the potential mechanism. Researchers found that thymol reduces the abundance of the detrimental bacteria *E. coli* in the GIT of pigs [43]. In another study, feeding pigs with microencapsulated thymol promoted the proliferation of beneficial bacteria in the colon and decreased the colonization of detrimental bacteria, such as Escherichia, Campylobacter, Treponema, and Streptococcus [106]. In chickens infected with C. perfringens, thymol inhibited C. perfringens proliferation and then alleviated intestinal damage and mortality [50]. In these studies, thymol also promoted the colonization of beneficial bacteria, such as Clostridium, Lactobacillus, and Bacteroides, to improve gut health. Additionally, thymol also exhibited a direct anti-bacterial effect inhibiting human pathogens, such as *E. coli*, Listeria monocytogenes, and C. perfringens, which are harmful to the body’s health [107,108,109,110].

Overall, thyme exhibits therapeutic potential to treat IBD through multiple mechanisms, including the intestinal barrier, inflammation, redox homeostasis, and gut microbiota. The deregulation of any one aspect influences others, which also renders IBD a complex and multifactor disease. As mentioned, a compromised gut barrier leads to the invasion of pathogens, leading to an imbalance in the gut microbiota and the activation of an immune response; uncontrolled inflammation causes enterocyte death and further aggravates intestine damage. Moreover, prolonged inflammation also leads to the accumulation of free radicals and imbalance of redox homeostasis, which exacerbate inflammation. Therefore, it is difficult to conclude which mechanism plays a pivotal role in mediating thymol action unless a dedicated molecular mechanism is illustrated.

## 7. Conclusions

Multiple factors, including the environment, microorganisms, and genetics, interact to promote the development and occurrence of IBD [112]. Environmental factors, such as diet, which lead to gut microbiota dysbiosis, alter host mucosal defenses, and induce intestinal OS, are deemed to be major potential risk factors for IBD [113]. Simultaneously, diet is also a double-edged sword that is beneficial to the prevention and relief of IBD. More recently, green tea polyphenols have been shown to benefit IBD patients by reversing gut dysbiosis [114]. Therefore, nutritional intervention should be further promoted as a low-cost, side-effect-free method for the prevention and treatment of IBD [115]. Thymol, a natural product derived from medicinal plants or herbs, is commonly used as a dietary supplement to prevent IBD by improving gut integrity, reducing gut OS, and modulating gut immune responses. In addition, thymol maintains a good intestinal microenvironment through its anti-bacterial properties. Currently, considering that the common drugs used for IBD, such as corticosteroids, mesalazine, and balasaladin, have relatively large side effects, herbal plant extracts, such as thymol, as natural anti-oxidants may represent promising substances in the complementary therapy of IBD, with special emphasis on prevention. However, there are still doubts regarding how thymol regulates mucosal immunity and the mechanism of treating IBD by improving OS in the intestinal tract. Meanwhile, there have been no clinical trials yet, so an effective method and dosage for people ingesting thymol need to be further studied. In conclusion, considering the health benefits of thymol and its non-toxicity, thymol is a potential drug and rational strategy for the relief of IBD.

## Figures and Tables

**Figure 1 antioxidants-11-01947-f001:**
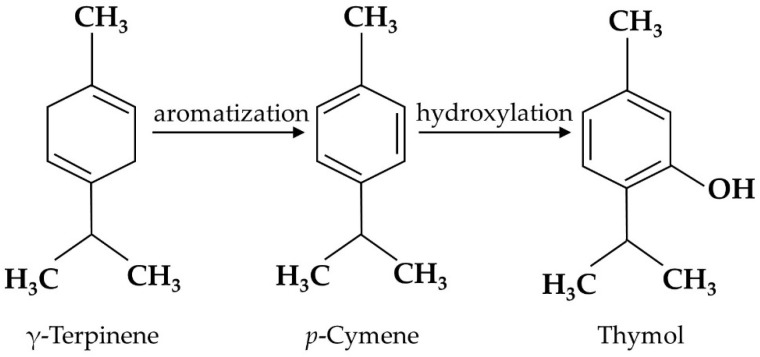
Thymol biosynthesis in plants. Thymol is synthesized by the hydroxylation of *p*-cymene, which originates from the aromatization of γ-terpinene.

**Figure 2 antioxidants-11-01947-f002:**
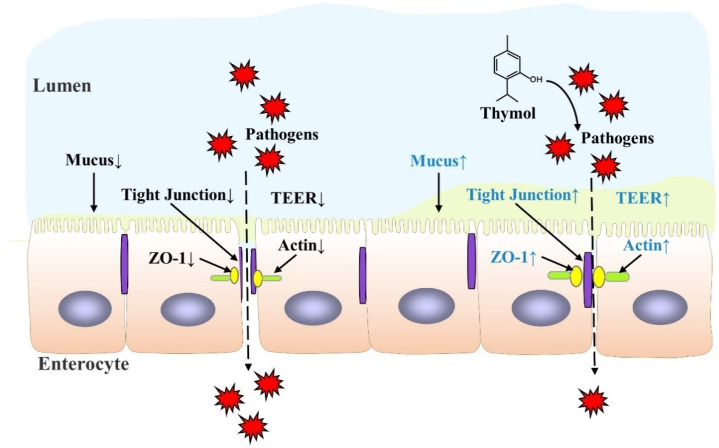
Action of thymol to protect intestinal barrier function. Pathogen infection in the intestinal lumen reduces mucus secretion and the expressions of tight junction proteins, and increases intestinal permeability, resulting in a “leaky gut”, which increases the risk of intestinal disease. Thymol has been shown to defend against pathogen invasion, promote mucus secretion, and enhance intestinal barrier integrity.

**Figure 3 antioxidants-11-01947-f003:**
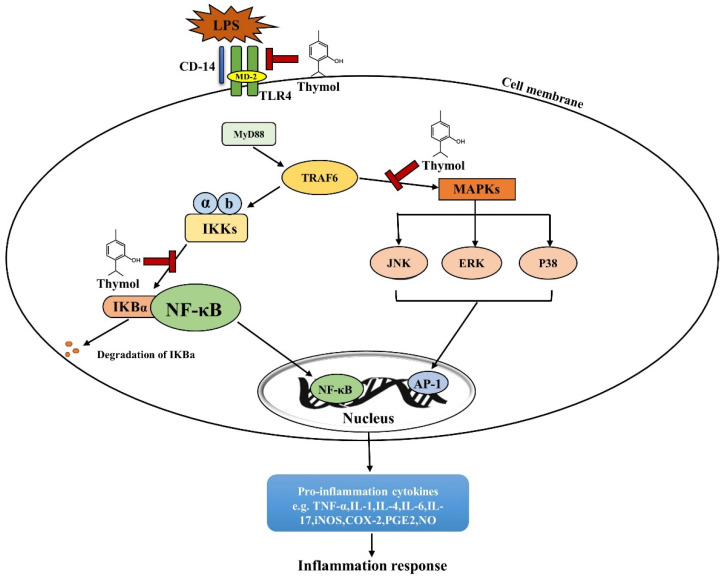
Mechanism of thymol in relieving inflammation. Cells receive various stimuli and activate IκB kinases (IκKs) through the TLR4 signaling pathway, and then IκB proteins are phosphorylated, ubiquitinated, and degraded, and NF-κB dimers are released. The NF-κB dimer is then further activated through various post-translational modifications and translocated into the nucleus to bind to target genes to promote pro-inflammatory gene transcription. In addition, by activating the MAPKs family, it can promote the expression of c-jun terminal kinases (JNK) 1/2, extracellular signal-regulated kinase (ERK), and p38 for activation, thereby promoting the nuclear entry of activator protein 1 (AP-1) to regulate the expression of pro-inflammatory genes. Thymol inhibits the dissociation of the IκB protein and NF-κB dimer and the activation of the mitogen-activated protein kinase (MAPK) signaling pathway to relieve inflammation.

**Figure 4 antioxidants-11-01947-f004:**
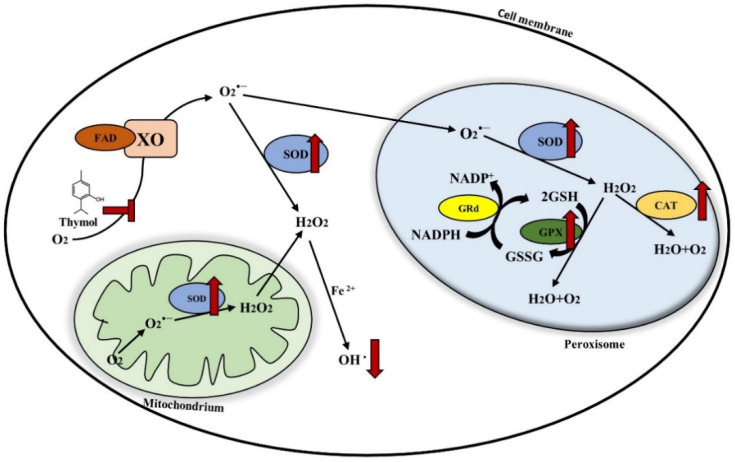
Mechanism of thymol scavenging free radicals. XO catalyzes the oxidation of xanthine and hypoxanthine with increased production of O_2_^•−^ in purine metabolism or O_2_^•−^ is generated in the mitochondrial inner membrane through the respiratory electron transport chain. The former enters the peroxisome and the latter is catalyzed by SOD in the mitochondria to generate H_2_O_2_. CAT and GPX in the peroxisome catalyze H_2_O_2_ to generate H_2_O and O_2_, and H_2_O_2_ in the cytoplasm is affected by Fe^2+^ to generate OH·. Elevated levels of free radicals can lead to oxidative stress. Thymol reduces ROS by directly scavenging free radicals, combining with XO, or increasing anti-oxidant enzyme activity.

**Table 1 antioxidants-11-01947-t001:** Protective function of thymol on the intestinal barrier.

Model	Thymol Dose	Effects	Ref.
In vivo			
Weaned pigs	50 mg/kg diet	Enrichment of 120 and 59 gene sets in oxyntic and pyloric mucosa↓	[29]
Ethanol-induced acute ulcer	10–100 mg/kg diet	Mucosal damage↓ Amount of mucus↑	[47]
Chicken infected with *Clostridium perfringens*	30 mg/kg diet	Intestinal lesions and mortality↓ *Lactobacillus salivarius* and *L. johnsonii*↓ *L. crispatus*, *L. agilis*, and *Escherichia coli*↑	[50]
Weaned pigs	100 mg/kg diet	Expression of ZO-1 and occludins in jejunal mucosa↓ *Enterococcus* genus and *E. coli*↓ Plasma diamine oxidase concentration↓ Weaning-induced intestinal OS↓	[43]
In vitro			
IPEC-J2	50 μM	TEER↑ Cell permeability↓ ZO-1and actin staining↑	[48]
Caco-2 cells	15 mg/L	COX1 transcription↑ COX1:COX2 ratio↑ TEER↑	[49]

↑ indicates for rising; ↓ indicates for descending.

**Table 2 antioxidants-11-01947-t002:** Summary of the effect of thymol in different inflammation models.

Model	Thymol Dose	Effects	Ref.
In vitro			
Mouse macrophages challenged with LPS	20 mg/mL	IL-1β expression↓	[59]
Mouse mammary epithelial cells challenged with LPS	10–40 μg/mL	TNF-α, IL-6, iNOS, and COX-2 expression↓ Phosphorylation of IκBα, NF-κBp65↓	[64]
Human peritoneal mesothelial cell line challenged with LPS	10–40 μg/mL	TLR4 expression↓ NF-κB p65, IκK, and IκBα phosphorylation↓ TNF-α, IL-6 expression↓	[65]
IPEC-J2 cells challenged with LPS	10–100 µM	IL-8 secretion↓	[66]
Chitin-induced airway epithelial cells	30 µg/mL	TLR4 is inhibited IL-25 and IL-33 release↓	[67]
HaCaT cells challenged with *Staphylococcus aureus*	512 µg/mL	IL-1β, IL-6, and IL-8 expression↓ Phosphorylation of p65 and IκBα↓	[68]
In vivo			
DSS-induced mouse colitis	30–60 mg/kg diet	NO, TNF-α, IL-1β, IL-6 expression↓ Phosphorylation of IκBα and NF-κBp65↓	[62]
Indomethacin-induced rat gastric ulcer	75–500 mg/kg diet	TNF-α, iNOS levels↓	[17]
Indomethacin-induced rat gastric ulcer	50–500 mg/kg diet	ROS, eNOS, TNF-α, caspase-3 levels↓ Prostaglandin E2 (PGE2) levels↑	[69]
Rat ulcerative colitis	100 mg/kg diet	COX-2, IL-6, IL-1β and TNF-α expression↓ mRNA level of NF-κB p65↓ Myeloperoxidase (MPO) activity, NO, and malondialdehyde (MDA) level↓	[63]
Ovalbumin-induced rat allergic rhinitis	20 mg/kg diet	Plasma IL-5, IL-13, IgE levels↓ TNF-α expression in the nasal mucosa of rats↓	[70]
*Clostridium perfringens* infection-induced chicken necrotic enteritis	15–60 mg/kg diet	TLR2 and TNF-α level in ileum↓	[18]

↑ indicates for rising; ↓ indicates for descending.

**Table 3 antioxidants-11-01947-t003:** Anti-oxidant properties of thymol.

Model	Thymol Dose	Response	Ref.
In vitro			
*Tert*-butyl hydroperoxide-induced OS in Chang cells	12.5–50 µg/mL	ROS generation and MDA level↓ Glutathione (GSH) level↑	[89]
Radiation-induced cytotoxicity in lung fibroblast (V79) cells	0–100 µg/mL	Radiation-induced lipid peroxidation↓ GSH, catalase (CAT), and superoxide dismutase (SOD)↑	[90]
Chinese hamster lung fibroblast cells (V79)	0–100 µg/mL	Radiation-induced genotoxicity and apoptosis↓	[91]
*Candida albicans*	5–20 µg/mL	Activity of CAT, glutathione peroxidase (GPX)↑	[92]
in vivo			
Imidacloprid-induced testicular toxicity	30 mg/kg diet	CAT, SOD, and GSH↑ MDA↓	[93]
L-arginine-induced acute pancreatitis	50–100 mg/kg diet	MPO and O_2_^•−^↓	[94]
High fat diet-induced obesity in mice	14 mg/kg diet	SOD and CAT activity in serum↑ MDA level in serum↓	[95]
Ovalbumin-induced asthma in mice	8–32 mg/kg diet	MDA level↓	[39]
Wistar rats	42.5 mg/kg diet	GSH-Px activity↑ Brain total anti-oxidant status↑ Proportions of docosahexaenoic acid (DHA)↑	[96]
Rats	10–20 mg/kg diet	MDA levels in testicles, liver, and kidney↓ Oxidative damage↓ Anti-oxidant levels and GSH levels↑	[97]
Cisplatin-induced nephrotoxicity in rats	20 mg/kg diet	GSH, SOD, and CAT levels in kidney↑ Caspase-3 and MDA levels in kidney↓	[98]

↑ indicates for rising; ↓ indicates for descending.

**Table 4 antioxidants-11-01947-t004:** Thymol prevents pathogen infection.

Model	Thymol Dose	Effects	Ref.
In vivo			
Weaning pigs	2% thymol diet	Colon probiotic bacteria↑ Potential pathogens↓	[106]
Cobb broilers	150 mg/kg diet	*Clostridium*, *Bacteroides*, *Lactobacillus*↑ Proteobacteria↓	[111]
Pathogen culture medium			
Ground chicken	100–200 ppm	*E. coli* O157:H7↓	[108]
Meat medium	46.875–6000 μg/mL	*E. coli, Salmonella*, and *C. perfringens*↓	[107]
Apple cider and milk	0.5 g/L	*E. coli* O157:H7 and *L. monocytogenes*↓	[109]
Cattle waste	6.7 mM	Fecal coliforms↓	[110]

↑ indicates for rising; ↓ indicates for descending.

## Data Availability

Not applicable.

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
