# Peer review of "Protective Effects of Natural Antioxidants on Inflammatory Bowel Disease: Thymol and Its Pharmacological Properties"

_antioxidants, 2022, doi:10.3390/antiox11101947_

Round 1

Reviewer 1 Report

The proposed review is intersting and well organized, offering important informations about the pharmacology of thymol and the possible use of it for IBD patients.

I suggest ro revise the introduction and the discussion discussing more in depth the role of the intestinal microbes in the pathogenesis of IBD, also considering the antimicrobial activity of thymol. I suggest to read and discuss at least the following papers:

Tomasello et al. doi: 10.3748/wjg.v20.i48.18121.

Sundin et al. doi: 10.1097/PSY.0000000000000470.

Nishida et al. doi: 10.1007/s12328-017-0813-5.

Santacroce et al. doi: 10.52586/4930.

Plant names must respect taxonomy rules: please revise l. 67-70

The legend of fig. 1 seems to be uncorrect, please revise

L. 80: which mucosa? Any mucosa, or a specific one?

Bacterial taxonomy requires italics for the name of the organisms (i.e., l. 112 and tab. 1 and 2)

Finally, revise the text for a number of typos.

Author Response

Dear Reviewers of Antioxidants,

Thanks very much for taking your time to review this manuscript. I really appreciate all your comments and suggestions! Please find my itemized responses in below and my revisions/corrections in the re-submitted files. Thanks again!

Major: I suggest revising the introduction and the discussion discussing more in depth the role of the intestinal microbes in the pathogenesis of IBD, also considering the antimicrobial activity of thymol. I suggest reading and discussing at least the following papers:

Tomasello et al. doi: 10.3748/wjg.v20.i48.18121;

Sundin et al. doi: 10.1097/PSY.0000000000000470;

Nishida et al. doi: 10.1007/s12328-017-0813-5;

Santacroce et al. doi: 10.52586/4930.

Answer: Thanks for the suggestion. We described the important role of gut microbes in pathogenesis of IBD and discussed how thymol influences gut microbiota potentially alleviating intestinal diseases in Line 339-367. The latest publication as mentioned were discussed and included in the manuscript in Line 62 (Tomasello et al. doi: 10.3748/wjg. v20.i48.18121); Line 144 (Sundin et al. doi: 10.1097/PSY.000000000000 0470); Line 342 (Nishida et al. doi: 10.1007/s12328-017-0813-5); Line 343 (Santacroce et al. doi: 10.52586/4930).

Minor:

  1. Plant names must respect taxonomy rules: please revise L.67-70

Answer: The names of plants have been revised following taxonomy rule in Line 90-94.

  1. The legend of fig. 1 seems to be incorrect, please revise

Answer: The legend of Fig 1 was changed to “Figure 1. Thymol biosynthesis in plants. Thymol is synthesized by the hydroxylation of p-cymene, which derives from the aromatization of γ-terpinene.” Line99-100

  1. L.80: which mucosa? Any mucosa or a specific one?

Answer: Based on the publication (MariaNieddu et al. doi: 10.1016/j.carbpol.2013. 10.084), thymol is detected in mucosa of both small and large intestines. The manuscript was revised in Line110 to make more clearly.

  1. Bacterial taxonomy requires italics for the name of the organisms (i.e., l. 112 and tab. 1 and 2)

Answer: The names and format of bacterial taxonom

Reviewer 2 Report

Major points

 1)    The paper aimed to review putative mechanism by which thymol, a natural monoterpene phenolic compound, alleviate inflammatory bowel disease (IBD). They focused on restored 1) burrier function of enterocytes, 2) alleviated bowl inflammation induced by a combination of TLR4 and NF-κB, and 3) enhanced body's antioxidant capacity. However, they did not clearly mention whether these 1)-3) mechanisms are interacted with each other or independently act to improve IBD. Please clarify these points.

 2)    In in vitro, it is true that polyphenols including thymol act as direct antioxidants via the auto-oxidation of phenolic hydroxyl group leading to scavenging effect on ROS. However, in vivo, most scientists now believe that polyphenols likely act as indirect antioxidants. Please check the following two literatures discussing the polyphenol/antioxidant controversy.

 https://www.sciencedirect.com/science/article/abs/pii/S0003986115003689

 https://academic.oup.com/jn/article/141/5/989S/4689148

 Thus, the authors should reconsider this point of view.

 Minor points

1)    Many technical terms lack full spellings, e.g. lipopolysaccharide for LPS, dextran sulphate sodium for DSS, Toll-like receptor for TLR, pattern-recognition receptor for PRR, and so on. Even though the authors are familiar with these terms, they should be written in full spelling at their first appearance.

 2)    Tables

It is difficult to grasp what kinds of models were used in some references. For instance, “Indo-induced rats gastric ulcer” and “Indomethacin-induced gastric ulcer model” in Table2. What is “Indo” in the former and what kind of animal species was used in the latter, etc. Even though most of the readers guess “Indo” is indomethacin, the authors should be more careful to summarize the tables. In all in vivo models, was thymol administered by oral gavage?

3)    Line 192, as far as I know, DPPH, a stable radical compound, is not classified as ROS.

Author Response

Dear Reviewers of Antioxidants,

Thanks very much for taking your time to review this manuscript. I really appreciate all your comments and suggestions! Please find my itemized responses in below and my revisions/corrections in the re-submitted files. Thanks again!

Major:

1) The paper aimed to review putative mechanism by which thymol, a natural monoterpene phenolic compound, alleviate inflammatory bowel disease (IBD). They focused on restored 1) barrier function of enterocytes, 2) alleviated bowl inflammation induced by a combination of TLR4 and NF-κB, and 3) enhanced body's antioxidant capacity. However, they did not clearly mention whether these 1)-3) mechanisms are interacted with each other or independently act to improve IBD. Please clarify these points.

Answer: We agree that these three mechanisms work together to maintain gut health. Malfunctions in any one of the gut barrier, inflammation, oxidative stress, and gut microbiota can contribute to gut disease. As mentioned earlier, a compromised gut barrier leads to the invasion of pathogens, leading to an imbalance in the gut microbiota and the activation of an immune response; uncontrolled inflammation causes enterocytes death and further aggravates intestine damage. Moreover, prolonged inflammation also leads to accumulation of free radical and imbalance of redox homeostasis, which exacerbate inflammation. Anyway, gut barrier function, inflammation, oxidative stress, and gut microbes are complementary and integrated. IBD is a multifactorial disease, thus any factors interrupt one or more mechanisms may contribute to the pathogenesis. In the manuscript, the interaction among 4 mechanisms in the pathogenesis of IBD was discussed and clarified in Line 368-376.

2) In in vitro, it is true that polyphenols including thymol act as direct antioxidants via the auti-oxidation of phenolic hydroxyl group leading to scavenging effect on ROS. However, in vivo, most scientists now believe that polyphenols likely act as indirect antioxidants. Please check the following two literatures discussing the polyphenol/antioxidant controversy.

https://www.sciencedirect.com/science/article/abs/pii/S0003986115003689

https://academic.oup.com/jn/article/141/5/989S/4689148

Thus, the authors should reconsider this point of view.

Answer:We agree that thymol acts as both indirect and direct anti-oxidant to enhance the anti-oxidative capacity. As mentioned in Line 91–93, thymol is detected in different orgrans, thus thymol could scavenge ROS in these organs as direct anti-oxidants via the phenolic hydroxyl group. Thymol has been reported to modulate the activities of anti-oxidation and mitochondrial function related enzymes and signaling pathway. For example, thymol regulates activities of XO, ETC related enzymes, HO1 and NRF2 signaling pathways, thus indirectly reduce OS. Moreover, as mentioned in previous section, inflammation and oxidative stress are highly correlated. Thymol interacts with TLR4-NFκB signaling to alleviate inflammation indirectly reducing oxidative stress. In addition, thymol metabolites, mainly thymol glucuronide and thymol sulfate are also detected in different organs, and they have been involved in the regulation of redox homeostasis. Overall, the anti-oxidative effect of thymol is the consequence of different mechanisms. We also add a comment to this point in L286-299.

Minor points

1) Many technical terms lack full spellings, eg. lipopolysaccharide for LPS, dextran sulphate sodium for DSS, Toll-like receptor for TLR, pattern-recognition receptor for PRR, and so on. Even though the authors are familiar with these terms, they should be written in full spelling at their first appearance.

Answer:All abbreviations were carefully checked and defined. For example, DSS for dextran sulfate sodium (L157), LPS for lipids Polysaccharide (L165), Toll-like receptor for TLR (L201), pattern recognition receptor for PRR (L209).

2) Tables.It is difficult to grasp what kinds of models were used in some references. For instance, “Indo-induced rat gastric ulcer” and “Indomethacin-induced gastric ulcer model” in Table2. What is “Indo” in the former and what kind of animal species was used in the latter, etc. Even though most of the readers guess “Indo” is indomethacin, the authors should be more careful to summarize the tables. In all in vivo models, was thymol administered by oral gavage?

Answer:I checked the specific drug of Indo in "Indo causes gastric ulcer in rats" in Table 2. According to the original article, the full article is only expressed by the abbreviation of “Indo”. But according to "Indo, is one of the most commonly used NSAIDs with potential hepatotoxicity." in the article, we speculate that “Indo” is the abbreviation of “Indomethacin”. In addition, the animal model used in the "Indomethacin-induced gastric ulcer model" is rats, which I will modify in more detail. Regarding the in vivo model covered in this paper, thymol was administered orally.

3)Line 192, as far as I know, DPPH, a stable radical compound, is not classified as ROS.

Answer: Removed DPPH as ROS in manuscript.

Reviewer 3 Report

The review of Liu and colleagues described the potential protective activity of thymol in inflammatory bowel disease (IBD) in terms of its effects on intestinal barrier function, inflammation-immune mediators, and oxidative stress. The following points are suggested of recommendation by this reviewer:

- Use of English. In general, it is recommended a careful revision of the use of English. Examples given at sentences: lines 27-29; 46-48; 102-103; 129-130; 196, 199. But, throughout the manuscript may also be suitable.

- Check the use of some abbreviation. OS for oxidative stress is not a well-known term.

- Introduction to IBD need a deep and critical revision, with more clear concepts, and also referring to the literature cited. Considering the topic of the paper, the following review work may fit well in some sections of the manuscript (DOI: 10.3390/ijms221910224). 

In lines 96-100, the introduction to IBD is again loose, and unconnected to the introduction in point 1. Recheck the information about IBD.

- Line 58. Is reference 1 (A genome-wide association study identifies IL23R as an inflammatory bowel disease gene. Science. 2006,), confirming this sentence? 

- Line 80, which organ mucosa?

- Lines 90-94. Ending the flow of this paragraph about pharmacokinetics with the last sentences seems inappropriate.

- Tables. Table 1 needs a deep revision of the writing. The effects need to be better explained and described with consistency. Also, consider revision of abbreviations.

- Lines 129-130. This sentence is a vague description

- General comment. Lack of studies about the effects of thymol in inflammation of immune cells, such as flow cytometry experiments of immune subsets, and cytokine production by T-cells, T-regulatory, B-cells, macrophages, etc. Also, there is no mention of studies evaluating the effect of thymol over gut microbiota and IBD.

- Lines 253-255. Diet (thymol) as the main way to prevent, and even, treat, IBD, and dietary supplement for the cure of IBD... this is not an appropriate conclusion. 

Author Response

Dear Reviewers of Antioxidants,

Thanks very much for taking your time to review this manuscript. I really appreciate all your comments and suggestions! Please find my itemized responses in below and my revisions/corrections in the re-submitted files. Thanks again!

  1. Use of English. In general, it is recommended a careful revision of the use of English. Examples given at sentences: lines 27-29; 46-48; 102-103; 129-130; 196, 199. But, throughout the manuscript may also be suitable.

Answer:

Line27-29: the statements of “It is not fatal but sometimes causes life-threatening complications, such as colon cancer.” were corrected as “It is not fatal, but sometimes causes life-threatening complications, such as colon cancer.”, Line 28-29;

Line46-48: the statements of “Currently, treatment is to manage inflammation in immune cells and reduce OS in intestinal epithelial cells to relief symptoms and prevent complication development for the remission of IBD, rather than a cure.” were corrected as “Recently, advances in dietary control of inflammation, OS and IBD incidence making dietary intervention are recommended as long-term prevention and therapy.”, Line 66-68;

Line102-103: the statements of “Thus, understanding how thymol protecting intestinal barrier function plays an important role in relieving IBD.” were corrected as “Thus, understanding how thymol protects the intestinal barrier plays an important role in relieving IBD.”, Line 148-149;

Line129-130: the statements of “Thymol is effective in restoring intestinal integrity, such as promoting mucus secretion, etc., to relieve intestinal diseases.” were corrected as “Thymol, through its antibacterial effect, defends against harmful bacterial invasion, promotes mucus secretion, and the expression of genes related to tight junctions in the intestinal epithelium can effectively restore intestinal integrity to alleviate intestinal diseases.”, Line 180-182

Line196: the statements of “In intestine of IBD patient, over production of ROS destroys cytoskeletal” were corrected as “In the intestine of IBD patients, overproduction of ROS destroys”, Line 271

Line199: the statements of “In addition, oxidative stress inflames gastrointestinal tract, and then lead to IBD disease.” were corrected as “In addition, OS inflames the gastrointestinal tract and further leads to IBD disease.”, Line 274

  1. Check the use of some abbreviation. OS for oxidative stress is not a well-known term.

Answer:Thanks for the suggestion, it is true that oxidative stress (OS) is not a well-known term, but we have a clear definition of OS as oxidative stress at Line 56 of the manuscript.

  1. Introduction to IBD need a deep and critical revision, with more clear concepts, and also referring to the literature cited. Considering the topic of the paper, the following review work may fit well in some sections of the manuscript (DOI: 10.3390/ijms22191 0224).

Answer:Anyway, thank you for your comments. The literature you provided is of great value, and I am citing it in Line 161-163: "The intestinal mucus layer is the first line of defense against gut bacteria, maintaining bacterial symbiosis with the host and preventing bacterial penetration into epithelial cells.". In addition, I Added thymol effects on gut microbes in the pathogenesis of IBD through its pharmacological properties such as antibacterial effects. Line319-344.

  1. In lines 96-100, the introduction to IBD is again loose, and unconnected to the introduction in point 1. Recheck the information about IBD.

Answer: Thank you for your opinion. I replaced the content of Line96-100 with Line141-144: “The intestinal barrier, composed of intestinal epithelial cells (IECs) and innate immune cells, maintains the balance between luminal contents and the mucosa. The intestinal mucosa and the contents of the intestinal lumen are in a state of func-tional balance, and the disturbance of this balance can lead to diseases such as IBD.”

  1. Line 58. Is reference 1 (A genome-wide association study identifies IL23R as an inflammatory bowel disease gene. Science. 2006,), confirming this sentence?

Answer:Thanks for your kindly pointing out the error, the reference inserted by Line 58 is indeed wrong and I have changed it (DOI: 10.1186/s40104-016-0079-7). Line 80

  1. Line 80, which organ mucosa?

Answer:Based on the publication (MariaNieddu et al. doi: 10.1016/j.carbpol.2013. 10.084), thymol is detected in mucosa of both small and large intestines. The manuscript was revised in Line 108-110 to make more clearly.

  1. Line 90-94. Ending the flow of this paragraph about pharmacokinetics with the last sentences seems inappropriate.

Answer:It's a good opinion! I think it would be more adequate to change the subparagraph to "Pharmacokinetic and pharmacological properties of thymol". Thus, I enriched this section, especially the pharmacological properties in Line106-120. And, I revised the last sentence to " This review focused the therapeutic potential of thymol in IBD due to its pharmacological properties, the mechanisms of which are detailed below." Line 127-138.

  1. Tables. Table 1 needs a deep revision of the writing. The effects need to be better explained and described with consistency. Also, consider revision of abbreviations.

Answer:I have revised Table 1 and expanded the description of the meaning in the table. Also, I double checked the suggestions for the acronyms you mentioned and adjusted them in time.

  1. Lines 129-130. This sentence is a vague description

Answer:I have changed your reference to Line129-130: “Thymol is effective in restoring intestinal integrity, such as promoting mucus secretion, etc., to relieve intestinal diseases.” I have modified it to Line180-182: “Thymol, through its antibacterial effect, defends against harmful bacterial invasion, promotes mucus secretion, and the expression of genes related to tight junctions in the intestinal epithelium can effectively restore intestinal integrity to alleviate intestinal diseases.”

  1. General comment. Lack of studies about the effects of thymol in inflammation of immune cells, such as flow cytometry experiments of immune subsets, and cytokine production by T-cells, T-regulatory, B-cells, macrophages, etc. Also, there is no mention of studies evaluating the effect of thymol over gut microbiota and IBD.

Answer:Thank you very much for your opinion! This review cites studies of thymol on macrophages Line108-109. We have added a review of thymol on cells such as T cells and T regulatory cells Line247-262. Currently, there is no literature report on the effect of thymol on B cells. Therefore, we hope that this review will pave the way for more research on the anti-inflammatory effects of thymol. At the same time, based on your suggestion, we have added the possibility that thymol can alleviate IBD by regulating the intestinal flora. Line 340-367.

  1. Lines 253-255. Diet (thymol) as the main way to prevent, and even, treat, IBD, and dietary supplement for the cure of IBD. this is not an appropriate conclusion.

Answer:I agree with your opinion. Although thymol has many pharmacological properties, such as anti-inflammatory, antioxidant, antibacterial and other effects, its pharmacological effects have not reached the level of being able to treat IBD. In addition, thymol has great pharmacological value, which is mainly reflected in prevention, and it will not be the main method for treating IBD. So, I changed this sentence to "Thymol, a natural product derived from medicinal plants or herbs, is used as a dietary supplement for the prevention and relief ofthe cure of IBD by improving intestinal integrity, OS and regulating immune response to reduce intestinal injury." Line 387-389

Round 2

Reviewer 2 Report

The authors tried to put too much information on the thymol’s mode of action against IBD. Consequently, important points of view on the protective effect of thymol on IBD became vague, which will make readers confused. E.c., regarding effects of thymol on gut microbiota, what kind of imbalance is observed in IBD and how thymol can improve it are not clearly discussed, even though the authors showed possible involvement of E. coli in the mode of action of thymol. In addition, there are so many grammatical mistakes in English.

Author Response

Dear Reviewers of Antioxidants,

Thank you very much for taking the time again to review this manuscript. I really appreciate all your comments and suggestions! They are very precious to me! Please find my itemized responses below and my revisions/corrections in the resubmitted files. Thanks again!

The authors tried to put too much information on the thymol’s mode of action against IBD. Consequently, important points of view on the protective effect of thymol on IBD became vague, which will make readers confused. E.c., regarding effects of thymol on gut microbiota, what kind of imbalance is observed in IBD and how thymol can improve it are not clearly discussed, even though the authors showed possible involvement of E. coli in the mode of action of thymol. In addition, there are so many grammatical mistakes in English.

Answer: Thank you for your opinion, your opinion is very pertinent! We consider that thyme exhibits the therapeutical potential to treat IBD through multiple mechanisms, including intestinal barrier, inflammation, redox homeostasis, and gut microbiota. Indeed, deregulation of any one aspect influences others, which also makes IBD a complex and multifactor disease. As mentioned, a compromised gut barrier leads to the invasion of pathogens, leading to an imbalance in the gut microbiota and the activation of an immune response; uncontrolled inflammation causes enterocytes death and further aggravates intestine damage. Moreover, prolonged inflammation also leads to accumulation of free radical and imbalance of redox homeostasis, which exacerbate inflammation. Therefore, it is difficult to conclude which mechanism plays a pivotal role in mediating thymol action unless a dedicated molecular mechanism is illustrated.

Regarding the effects of thymol on the gut microbiota, we have revised at line265-291: “Mammalian gut microbes are mainly composed of four phyla, including Firmicutes, Bacteroidetes, Proteobacteria, and Actinobacteria. Alterations in the gut microbiota are observed in patients with IBD compared with health individuals. Notably, the abundance of phylum Firmucutes is reduced in stool of UC patients [104]. Although there is a lack of clinic research about thymol modulated gut microbial structure in IBD, multiple animal studies have described thymol increases Firmicutes proportional abundance potentially against IBD pathogenesis [105]. However, a direct causation be-tween IBD and dysbiosis has not been clearly established in human. The change in gut microbes by thymol as a therapeutical mechanism requires more elucidative studies. Currently, studies of microbial etiology of IBD indicated the persistent pathogen infection, such as enterotoxic E. coli, causes “leaky gut” and chronic inflammation, subsequently leads to the excessive translocation of intestinal bacteria and the dysbiosis between “beneficial” and “detrimental” bacteria, resulting in IBD [106]. The members of the Proteobacteria phylum, notably Enterobacteriaceae E. coli, are increased in IBD patients relative to healthy individuals [107]. Several studies indicate the anti-pathogen capacity of thymol in different animal infection model and bacterial culture, as illustrated in Table 4, as the potential mechanism. Researchers found that thymol reduces the abundance of detrimental bacteria E. coli, in the GIT of pigs [43]. In another study, feeding thymol supplied to pigs with microcapsules promotes the proliferation of beneficial bacteria in the colon and decreases the colonization of detrimental bacteria such as Escherichia, Campylobacter, Treponema, and Streptococcus [108]. In Chicken infected with C. perfringens infected chicken, thymol inhibits C. perfringens proliferation and then alleviates intestinal damage and mortality [109]. In these studies, thymol also promotes the colonization of beneficial bacteria such as Clostridium, Lactobacillus and Bacteroides to improve gut health. Additionally, thymol also exhibits direct anti-bacterial effect inhibiting human pathogens such as E. coli, L. monocytogenes, S. aureus, and C. perfringens, in different culture medium [110-113].”.

In additon, we apologize for the poor language in our manuscript. We have worked on the manuscript for a long time, and repeated additions and deletions of sentences and chapters clearly contributed to poor readability. We have now made extensive language and readability changes throughout the article. We sincerely hope that there has been a substantial improvement in the language level.

Reviewer 3 Report

The authors have successfully addressed most of the comments raised by previous reviewers

Author Response

Dear Reviewers of Antioxidants,

Thank you very much for taking the time to review this manuscript. I really appreciate all your comments and suggestions! They are very precious to me. Thanks again!

The authors have successfully addressed most of the comments raised by previous reviewers

Answer: Thank you for your valuable comments earlier, it's been invaluable to me! Thank you again!
